# *Salmonella* Surveillance Systems in Swine and Humans in Spain: A Review

**DOI:** 10.3390/vetsci6010020

**Published:** 2019-02-20

**Authors:** Marta Martínez-Avilés, Macarena Garrido-Estepa, Julio Álvarez, Ana de la Torre

**Affiliations:** 1Animal Health Research Center (INIA-CISA), Ctra Algete a El Casar s/n, 28130, Valdeolmos, 28040 Madrid, Spain; garrido.macarena@inia.es (M.G.-E.); torre@inia.es (A.d.l.T.); 2VISAVET Health Surveillance Center, Complutense University of Madrid, 28040 Madrid, Spain; jalvarez@visavet.ucm.es; 3Animal Health Department, Veterinary School, Complutense University of Madrid, 28040 Madrid, Spain

**Keywords:** zoonoses, foodborne, disease control, public health, domestic livestock, pig, one health

## Abstract

Non-typhoid salmonellosis is a common and problematic foodborne zoonotic disease in which pork and pork products can be an important potential source of infection. To prevent this disease, important efforts to monitor the situation in the main source, livestock, are conducted in most developed countries. In the European Union, European Food Safety Agency (EFSA) and European Center for Disease Control (ECDC) compile information at the member-state level, even though important differences in production systems and surveillance systems exist. Here, *Salmonella* surveillance systems in one of the main sources of foodborne salmonellosis, swine, and humans in Spain were reviewed to identify potential gaps and discuss potential ways of integration under a “One-Health” approach. Despite the extensive information generated through the surveillance activities, source attribution can be only routinely performed through ad-hoc outbreak investigations, and national reports on human outbreaks do not provide sufficiently detailed information to gain a better understanding of the epidemiology of the pathogen. Human and animal monitoring of *Salmonella* would benefit from a better exchange of information and collaboration. Analysis of spatio-temporal trends in livestock and humans could help to identify likely sources of infection and to target surveillance efforts in areas with higher prevalence or where specific strains are found.

## 1. Introduction

*Salmonella* is a ubiquitous genus of bacteria commonly found in the intestines of healthy birds, reptiles and mammals that can cause one of the most common foodborne illness in humans [1]. According to the European Food Safety Agency (EFSA), the main species in the genus, *Salmonella enterica*, is one of the top agents involved in foodborne outbreaks in Europe including Spain, even though disease burden is likely severely underestimated because infection can be asymptomatic or not sufficiently severe to prompt testing [2]. Additionally, non-severe cases can be treated without further investigation of the subtype. 

Poultry *Salmonella* is under an official control program in the European Union (EU) since 2004. Consequently, there was a significant reduction of *Salmonella* in humans and poultry during the period 2008–2016 [3], particularly due to *S.* Enteritidis. In contrast, there was an apparent increase of notifications of *S.* Typhimurium cases, which are less likely associated with the consumption of eggs and egg products and more predominantly found in pork and pork products [4]. 

Spain is one of the main swine-producing countries, currently ranking first in number of swine in the EU (with 28.3 million animals in 2015) [5]. The production of pork in 2015 reached 3.8 million tons, with more than 45 million animals being slaughtered. Worldwide, Spain is the fourth largest pork producer after China, United States, and Germany. Mainly an exporting country, Spain has also become the EU’s third largest exporter of swine after Germany and Denmark. The swine industry accounts for 14% of the final agricultural production in Spain, and it is the most important livestock species in economic terms, representing 37% of the final livestock production. 

Pork is the main source of human salmonellosis after poultry in the EU [6], given that it is the third most frequently contaminated meat after fresh chicken and turkey [7], and it is widely consumed. Because of this, monitoring and surveillance activities have been implemented along the food chain to assess the risk posed by pork and pork products as a source of *Salmonella* for the general public and to prevent outbreaks. In addition, the threat posed by the increasing occurrence of infections caused by antimicrobial resistant *Salmonella* strains in humans is another reason to perform “One-Health” surveillance efforts to control *Salmonella* at its source [8].

Swine can acquire *Salmonella* infection from a contaminated environment or feed, or through direct contact with infected animals. Infected pigs can remain carriers of *Salmonella* and shed the bacteria via the feces intermittently for many months [9]. There is a risk of cross-contamination of carcasses with feces of infected or carrier animals at slaughter. Prevention of *Salmonella* infection in pigs at the farm is performed through the regular monitoring of pig feed and implementation of basic biosecurity measures and in certain cases vaccination. However, environmental persistence, high turnover of young stock and incoming replacement stock pose significant barriers to eliminate *Salmonella* at the farm. The *Salmonella* status of a pig can be monitored through serological tests performed on meat-juice or serum samples [10] or, more frequently, through the bacteriological analysis of feces collected at either the farm or the slaughterhouse, where mesenteric lymph nodes can be also collected [7]. There are also microbiological tests conducted on meat and carcasses to verify hygiene practices through the food chain, but in the case of positive results it is often not possible to establish whether the tissue contamination originated from an infected pig farm or occurred as a result of a contaminated environment [11] or insufficient hygiene practices during meat processing [12].

Spain is divided into 19 regions. Health competencies are transferred to the regions, who report to the national authorities. Animal health surveillance is the responsibility of the Regional Departments and the Ministry of Agriculture. However, meat inspection falls under the responsibility of the Regional Departments and Ministry of Health, together with human disease surveillance. Since 2004, EFSA has analyzed comparable data on zoonotic foodborne diseases from all EU Member States and harmonized prevalence targets are set. Member States inform EFSA and the European Center for Disease Control (ECDC), in agreement with the European Commission Directive 2003/99/EC, of the results of the monitoring systems in place (compulsory or voluntary monitoring programs, surveys, other procedures of sampling and lab reports), which is published in an Annual EU Summary report.

The EU-funded Joint Research Project “NOVA” (Novel approaches for design and evaluation of cost-effective surveillance across the foodchain) [13] under the Horizon 2020 grant agreement “One-Health”—European Joint Program (EJP) number 773830—seeks to develop new surveillance tools and methods and aims to harmonize and optimize the use of existing surveillance data on zoonotic foodborne diseases. Under this context, we review and describe the *Salmonella* surveillance systems in Spain “from farm to fork”, that is, from its animal source (here swine) up to the identification of *Salmonella* of animal origin in human outbreaks. Our aim is to identify potential gaps and to assess the feasibility of a more integrated approach in a “One-Health” framework. We argue the differences between systems and potential ways to integrate surveillance across sectors if needed for the benefit of all parties. 

## 2. Search and Review Strategy

We performed a non-systematic review on the ongoing monitoring programs for *Salmonella* in the animal reservoir (swine) and in humans across health institutions in Spain. Since the scope of the paper is limited to Spain, a list of the institutions’ and systems’ acronyms used (mostly of Spanish origin) is provided in Table 1.

Surveillance of *Salmonella* in swine is coordinated by MAPA, and laboratory testing is conducted in the national and regional official laboratories and MAPA-approved animal health laboratories, including VISAVET. Official sources of information on monitoring in swine are listed in the MAPA website dedicated to antimicrobial resistance and zoonosis surveillance (Table 2), which includes legislation and links to MAPA reports (2012–2016) and EFSA’s Spain reports (2004–2016) on zoonosis and antimicrobial resistance annual surveillance results. EFSA’s annual country reports include information on the reporting and monitoring system on zoonoses sent by a Reporting Officer from that country. MAPA (Animal Health Section) is the Reporting Officer from Spain, and gathers information from other National Reporters, namely AECOSAN, ISCIII, other sections within MAPA (Farm and Traceability Registries; Animal Feed), VISAVET, and the Regional Animal Health Departments. Here, we selected those reports that provided detailed information on the monitoring design and results of *Salmonella* in pigs to assess the potential uses of that information. In consequence, only EFSA annual country reports from 2007 to 2012 were selected. The information on human surveillance contained in the EFSA annual country reports is very brief. Therefore, we tracked the websites of other national institutions mentioned in EFSA’s Spain annual reports (ISCIII, AECOSAN) to retrieve further information on *Salmonella* surveillance in humans and in meat and meat products, respectively. Both ISCIII and AECOSAN belong to the Ministry of Health (MSSSI). Also, we completed our search by reading the pertinent legislation on *Salmonella* surveillance and by searching in PubMed and in the scientific bibliography archive of the Complutense University of Madrid scientific articles and doctoral thesis on *Salmonella* in pigs, particularly in Spain (combination of search terms “*Salmonella*”, “pigs”, “Spain”, “epidemiology”). This way, general information on *Salmonella* epidemiology was also retrieved. 

The Spanish National Epidemiological Surveillance Network (RENAVE) is responsible for the surveillance and control of infectious diseases in Spain. This network is composed by the 19 Spanish autonomous regions and it is coordinated by the MSSSI with the scientific and technical support of ISCIII. The Notifiable Diseases regional data is merged at ISCIII where the National Database is maintained. At ISCIII, the information is compiled, analyzed, and disseminated. One of the authors of this manuscript (M.G.-E.) worked for more than ten years at ISCIII and participated in the definition of the surveillance protocols for Notifiable Diseases in Spain. Protocols for each notifiable disease on use by the RENAVE are publicly available and were approved by the Inter-territorial Council of the National Health System (Table 2). Each protocol includes a general description of the epidemiology of the disease, standard definitions of cases classification, outbreak, and notification system and actions after a case or an outbreak have been confirmed. ISCIII holds three different databases: Notifiable Diseases (incidence on human salmonellosis that is part of this list only since 2015); Foodborne Outbreak Investigation Database (human salmonellosis outbreaks information since 2006); and an additional database created with the national microbiological laboratories data: SIM (identification and distribution of *Salmonella* serovars that are circulating in Spain reported voluntarily by participant laboratories since 1995). The results of the three databases are published in two annual reports (SIM reports; and RENAVE reports that combine Notifiable Diseases + Outbreak Investigation) available at the ISCIII website.

Tracking the legislation and the scientific literature [14,15], we reached the information gathered by the hospitals and primary health centers, which are the first locations where a suspect case is recorded. The information gathered by primary health centers and hospitals is available through the Statistics Portal of the MSSSI. Regional authorities can access these data and recapture any pending cases to notify them through RENAVE.

AECOSAN compiles and publishes the analysis results of *Salmonella* alerts in food through SCIRI, and then disseminates the results with an annual report available online. For this manuscript, we downloaded the reports corresponding to 2014 and 2015, because these were the years in which *Salmonella* cases started to be reported by the regional authorities to the RENAVE.

Based on the authors’ experience on epidemiological surveillance, we searched, across the existing monitoring systems in Spain on swine-related *Salmonella*, for information on sampling strategy, frequency of sampling and testing, information included in the dissemination and, most particularly, whether there was evidence of communication or actions relative to the results obtained with other stakeholders involved in *Salmonella* monitoring and results of any potential joint epidemiological analysis. For the “One-Health-EJP” project we also searched for evidence on spatial dissemination of results.

## 3. Findings

We first present the sources in which the information on *Salmonella* surveillance of swine origin in Spain can be found, followed by an analysis of the information contained in them at the animal, meat, and human source.

### 3.1. Sources of Information

Table 2 shows the websites visited for the non-systematic review of surveillance sources on *Salmonella* of swine origin of interest for Spain.

The legislation consulted is summarized in Table 3: (a) European (OJ = Official Journal) and (b) Spanish (BOE = State Official Bulletin, acronym in Spanish [Boletín Oficial del Estado]).

The information in the 11 scientific articles and 3 PhD theses selected from the literature was related to the results of point surveys, improved diagnostic methods, or influence of potential risk factors in Spain. Many referred the mentioned legislation which confirmed the coverage of the official sources of Salmonella monitoring in Spain. The articles and PhD theses selected are presented in the Appendix A.

### 3.2. Surveillance at the Pig Source

Routine surveillance of *Salmonella* in pigs in Spain is carried out at slaughter through the annual sampling of feces or lymph nodes of carcasses, in agreement with European Commission (EC) Regulation 2160/2003. The overall aim of the EC Regulation 2160/2003 is to reduce the incidence of *Salmonella* across the food chain through harmonized sampling schemes to obtain comparable prevalence estimates.

The slaughterhouses selected for sampling every year process at least 50–60% of the annual slaughter pigs, and slaughterhouses from at least 50% of regions in Spain are included (Figure 1). Sampling is then stratified by slaughterhouse based on its annual throughput, and one or more samples (typically fecal samples) are collected from a variable number of farms (ranging between 160 and 400 farms sampled annually in 2002–2015) and processed individually or as pools. 

In addition, in recent decades larger surveys have been conducted at the request of the EU. A large survey was conducted in slaughter pigs (Commission Decision 2006/668/EC) to establish the baseline prevalence of infection in the different member states of the EU. For this purpose, 2619 lymph node samples from pigs originating from different farms were collected in Spain from slaughterhouses accounting for >80% of the of all slaughtered fattening pigs.

Both data sources (annual monitoring program, EFSA baseline survey) revealed a prevalence of infection around 30% in slaughter pigs, though a wider range is observed when looking at the results from the annual program (generated through a smaller sample size) (Figure 1).

The serotypes with a higher average proportion of isolates from 2008 to 2013 in slaughter pigs were *S*.1,4, [5],12:i:- (9.08%), followed by *S*. Rissen (7.99%), *S*. Typhimurium (7.36%), *S*. 4,5,:i:- (6.06%) and *S*. Derby (5.09%). However, when disaggregating the slaughterhouse results by year of sampling (Figure 2), *S*. Typhimurium was the only serotype isolated every year, reaching the highest proportion of isolates in 2009 (11.66%). *S.* Rissen and *S*. Derby are the other two serotypes most frequently isolated in slaughterhouses, only missing in 2010. The highest proportion of *S.* Rissen (9.89%) also happened in 2009. *S.*1,4, [5],12:i:- was only isolated in 2012 and 2013 (8.59% in 2012 and 9.57% in 2013), and *S*.4,5,:i:- only in 2011. Curiously, *S.* Anatum was isolated from slaughterhouses only in 2008 and in 2011, and with a low proportion (<2%). 

A single national-level survey was conducted in 2008 to estimate the baseline prevalence of *Salmonella* infection in breeding farms as part of a larger effort at a European level (Commission Decision 2008/55/EC; EFSA, 2009). In Spain 359 holdings (150 breeding and 209 production holdings, sample size set to estimate a 50% expected prevalence with a 7.5% accuracy at a 95% confidence level) and 3,590 pens were sampled. A farm-level prevalence of 64.0 and 53.1% in breeding and production holdings was found, respectively, placing Spain as the country with the highest and second highest prevalence of *Salmonella* infection in each subpopulation [1]. The serotypes with a higher proportion of isolates in breeding farms were *S.* Rissen (15.88%), *S.* Typhimurium (13.09%), *S.* Anatum (9.47%) and *S.* Derby (8.08%).

### 3.3. Salmonella in Meat and Meat Products

*Salmonella* status in meat is monitored at the slaughterhouse through the analysis of tissue samples in swine carcasses (20 cm^2^ in 4 tissue samples using an abrasive sponge) and checks of the operator’s microbiological tests to control *Salmonella* through the food chain, in accordance with EC Regulations 217/2014, 218/2014, 1441/2007, and 2073/2005. If positive results are obtained, AECOSAN communicates it to MAPA so that further action can be pursued at the farm of origin, in particular regarding biosecurity practices and *Salmonella* testing. 

In Spain, a minimum of 50 carcasses are randomly sampled per year per slaughterhouse by the official veterinarian at the slaughterhouse, each one from a different farm (unless it is a small slaughterhouse, in which case the number of samples depends on the result of a risk assessment). If 3 or more samples are positive (out of the 50), corrective action is taken. The official veterinarian informs its region of the results of this sampling, together with the information about the number of positive samples obtained from the additional carcass sampling performed by slaughterhouse operators. The corrective action plan can include an investigation of the origin of the animals and the farm biosecurity measures, which implies a close collaboration with MAPA, which is the competent authority on animal health. Meat product testing is the responsibility of the business operator of slaughterhouses or establishments producing minced meat, meat preparations or mechanically separated meat, and is regulated by law (EC Regulation 20173/2005). This regulation allows the business operator to decide the sampling frequency according to its Hazard Analysis and Critical Control Points (HACCP)plan, but specifies a minimum requirement to guarantee a comparable level of control with other EU countries. This regulation also specifies the diagnostic test that should follow the standards published by the International Organization for Standardization (ISO) number 6579 for *Salmonella* or any other validated analytic method.

The information on pork meat monitoring contained in the annual country reports, provided to EFSA by the Regional Health Services, includes the number of units tested (each unit corresponds to 25 g of tissue sample) and the number that resulted positive to *Salmonella* in fresh meat (at the processing plant, at retail and at the slaughterhouse), in meat products raw but intended to be eaten cooked (at the processing plant and at retail) and occasionally, in raw but ready to be eaten raw meat products. The reports also specify the serotype when available. There is a low percentage of units positive to *Salmonella* (<5% of total units tested). Similar to slaughter pigs, *S*. Typhimurium has been detected every year (of the same period analyzed for pig status: 2008–2012), with the overall average prevalence of *S.* Typhimurium during this period being less than 2%. However, one cannot deduce from these data how many constituted a positive sample according to the regulation that led to corrective action. Surveillance at the pig and pig meat sources is summarized in Figure 3.

Table 4 shows the most relevant results on food monitoring provided by SCIRI from 2014 to 2016. During this period 7.75% of the overall alerts were related to *Salmonella* spp. presence in animal products; 0.39% of the information notifications were associated with *Salmonella* spp. biological risk in Spanish animal products; and 0.22% of the border rejections were related to *Salmonella* spp. presence in animal products. 

### 3.4. Cases and Outbreak Investigation in Humans

Figure 4 summarizes the surveillance of *Salmonella* in humans, which starts with the onset of clinical signs in a suspect patient.

Patients exhibiting diarrhea, fever, abdominal pain or vomiting who seek primary or specialist care at the local public outpatient health center or are admitted/referred to a hospital, are registered as “cases” if, in addition, *Salmonella enterica* (other than Typhi or Paratyphi) is confirmed in the laboratory by isolation in feces, clinical specimens (infected wound, etc.) or any sterile tissue / body fluid (blood, urine, etc.). An outbreak is defined as two or more *Salmonella* cases with a common source as background exposure. Besides the hospital and regional reference laboratories, the National Center of Microbiology, also at the ISCIII, is the Reference Laboratory for *Salmonella* and *Shigella* in Spain, and can provide deeper microbiological analyses than the other laboratories mentioned. ISCIII holds clinical information (symptoms onset and outcome), epidemiological data (outbreak association and outbreak identifier code) and microbiological variables (sample type and microbiological identification to serovar level). Information on hospitalized cases can be found in the hospitalizations Minimum Basic Data Set (MBDS) maintained since 1987 by the MSSSI. When *Salmonella* outbreaks are microbiologically confirmed to be of food origin, AECOSAN notifies it, when applicable, nationally through SCIRI and internationally through RASFF, ensuring the exchange of verified information and follow-up actions within a network of EU Member States. 

During the year 2014, 13 of the 19 Spanish regions (55.5% of the overall population in the country) reported salmonellosis cases (7,295 in total). The number of regions reporting increased in 2015, with 15 regions (70.4% of the overall population in the country) notifying 9,069 cases to ISCIII (Table 5). 

Table 6 shows the *Salmonella* spp. serovars distribution reported by RENAVE. 

The MBDS compiles information from more than 95% of the public/private hospitals and of 99% of the discharges in Spain, and in some regions 100%. Clinical information about severe cases of salmonellosis requiring hospitalization can be obtained from it. Hospitals confirm the cases in their own laboratory or by sending the sample to the national or the regional reference laboratories, but no detailed information of serotypes is provided in the dataset. The MBDS was launched in 1987, with constant improvement in the data collection. The last improvement was made in 2016 in which the dataset started to include ambulatory surgeries and procedures and switched to the 10th edition of the International Coding of Diseases (ICD-10) (Royal Decree 69/2015). Non-Typhi, non-Paratyphi *Salmonella* diagnosis are coded under the ICD-10 code A02. Overall, there were 3526 recorded hospitalizations related to non-Typhi, non-Paratyphi *Salmonella* in 2014 and in 2922 (82.9%) hospitalizations *Salmonella* infection was the main diagnosis. In 2015, there were 3776 hospitalizations related to non-Typhi, non-Paratyphi *Salmonella* and in 3185 (84.3%) hospitalizations *Salmonella* infection was the main diagnosis (Table 1).

The outbreaks dataset maintained by ISCIII contains further information about the outbreak including food item/s investigated. Outbreak information can also be linked to the individual patient’s information collected in the Notifiable Diseases database through the outbreak identifier code provided in both outbreaks and individual notifications. In 2014, 241 *Salmonella* associated outbreaks involving 1681 individuals were notified by the regions to ISCIII (Table 1). Among them, 249 (14.8%) were hospitalized and 7 (0.4%) died. Half of the outbreaks were caused by *S.* Enteritidis (53.9%), followed by *S.* Typhimurium (17%). In total, 73% (176/241) of the outbreaks were foodborne, one of the 176 transmitted by water. Some suspect food was identified in 72.7% of the outbreaks. Among suspect foods, the most commonly implicated food was the egg and its derivatives (68.8% of the outbreaks with food identification), followed at a great distance by meat and meat products (11.7%). In 2015, 281 *Salmonella* associated outbreaks involving 1920 individuals were notified (Table 1), 35 (1.8%) of them were hospitalized and 2 (0.1%) died. In 116 (77.3%) outbreaks the agent was *S.* Enteritidis, followed by *S.* Typhimurium with 30 (20%) outbreaks. Some suspect food was identified in 59.4% of the outbreaks and, among them, the most commonly implicated food was the egg and its derivatives (77.2% of the outbreaks with food identification). In 10.2% of the outbreaks the suspect food was meat and meat products.

SIM reports contain information on circulating *Salmonella* serovars. For all pathogens, SIM coverage of the Spanish population was 34% in 2014 and 30% in 2015. During 2014 and 2015, 72 microbiology laboratories of 11 Spanish regions participated in the SIM providing non-Typhi, non-Paratyphi *Salmonella* isolates information. In 2014 and 2015 respectively, 5001 and 5215 non-Typhi, non-Paratyphi *Salmonella* isolates were reported. 

#### Datasets Linkage

Table 7 shows the summary of the results from the different human *Salmonella* surveillance data sources by region.

Regions notifying cases at national level to the Notifiable Diseases system present a very similar number of reported cases to the positive isolates reported across SIM, showing that both systems are providing similar information. Some differences in numbers, especially when there are more positive isolates reported in the SIM, could be related to more than one isolate recovered from the same patient. The table also shows that the proportion of hospitalizations compared with the number of cases reported to the Notifiable Diseases system was 30.5% (4999 hospitalization vs. 16364 cases reported). This implies that most of the regions are reporting mild *Salmonella* cases attending primary care across both systems, Notifiable Diseases and SIM. It is expected that the four pending regions will join soon and report data from 2015 onwards to the Notifiable Diseases system. 

## 4. Discussion

This study reviews the information across the different reports and institutions participating in the surveillance of *Salmonella* in Spain. We have selected the swine reservoir given the importance of this livestock species in Spain and the lack of official control program for *Salmonella*, to connect the main stakeholders of surveillance in animals and humans in Spain, which are ultimately MAPA and MSSSI. The EFSA and the ECDC request EU Member States to report the design and results of foodborne zoonotic diseases surveillance by gathering such information directly from each of the stakeholders involved. The most extensive and detailed information in swine origin reports corresponds to the status of *Salmonella* at the animal source, where there seems to be a higher degree of collaboration between the MAPA and MSSSI than in the event of *Salmonella* detection in food items in human cases or outbreaks. The surveillance of *Salmonella* in pigs, meat, and humans are not comparable *per se*, since they are designed for different purposes. In pigs and in meat, the objective is to routinely monitor the burden of *Salmonella* infection to reduce the risk of human illness through foodborne contamination or from contact with infected pigs [16]. The risk reduction measures in pigs and meat differ, since meat contamination could be also a result of poor hygiene practices at slaughter or processing. In humans, the objective is to identify *Salmonella* in clinical suspects as soon as possible to be able to prevent new cases or outbreaks by the enforcement of hygienic practices and by withdrawing or treating the suspect source of infection [17]. 

Structurally, all surveillance systems on swine-related *Salmonella* include similar information on sampling strategy, types of samples and, frequency of sampling, laboratory testing, and results and dissemination. Under the prism of a collaborative “One-Health” surveillance system, there is, however, ample room for improvement. Bordier et al [18] propose 6 degrees of possible collaboration in a multi-sectoral surveillance system in the planning, data collection (sampling and lab testing), data sharing, results sharing, data analysis/interpretation, dissemination to decision makers and communication to surveillance actors and end users. After the analyses of the information in the different datasets and websites, we found very limited evidence of a high degree of collaboration in the steps proposed for a cooperative surveillance process. More likely, all steps are being taken separately for each sector, but the results are being shared periodically. The only link we found between sectors was in the procedures regarding a positive result in meat, but we could not find any report in which such collaboration was reflected. 

In addition, we found at least 7 different databases (4 from the human sector: SIM, Notifiable diseases, Outbreak investigation, and the hospitals’ MBDS; 1 from the meat sector: SCIRI-AECOSAN; and 2 from the veterinary sector: national monitoring, EFSA surveys) with information on *Salmonella* surveillance in Spain. Better knowledge of these databases to support an integrated “One-Health” surveillance approach could help to better focus the efforts in *Salmonella* control, for example by performing a joint analysis of the distribution and trends of *Salmonella* serovars in time and space and to determine more precisely the burden of each *Salmonella* serovar. For example, in Canada, weekly counts of *Salmonella* from farm animals, meat and humans allowed to create baseline models and identify significant clusters across the different sectors [19]. While identifying “hot-spots” of *Salmonella* specific serovar distribution would be desirable, the substantial differences in disease dynamics among *Salmonella* serotypes must be taken into account for a correct epidemiological interpretation of results. Arnedo-Pena et al [4] highlight some differential characteristics regarding the incubation period, outbreak duration, attack rate, hospitalization rate, probability of reaching a microbiological diagnosis, and human behavior that influence the variation in disease dynamics of foodborne outbreaks of *S.* Typhimurium and *S.* Enteritidis in Spain. It would be desirable to characterize also other serovars identified in animal health surveillance results (i.e., in pigs: *S.* Rissen and *S.* Derby) to target interventions. Human behavior is so predominant in the exposure to foodborne diseases that the inclusion of social sciences in the multidisciplinary approach to investigate *Salmonella* changes over time have already been suggested [8].

Linking outbreak investigation with farm status can be very time-consuming and an association with the sufficient level of confidence hard to obtain. So far, *Salmonella* in pigs is not under an official control program in Spain. Spain is classified as a “high prevalence” country according to EFSA. The monitoring of *Salmonella* in slaughter pigs is representative of the production system in Spain, in which more than 80% of the total census are fattening pigs [20]. The detailed census of pig holdings available in Spain at the coordinate level would in principle facilitate spatial analysis of the surveillance results, at least at the descriptive level. A deeper epidemiological analysis of the surveillance data is underway to assess their usefulness for identification of spatial and temporal trends in detection of *Salmonella*/specific strains, and for the early detection of emerging strains. 

*Salmonella* surveillance in meat and meat products is the responsibility of the business operator, and the MSSSI vet officially checks and oversees the surveillance. Again, we have not found any evidence of a joint epidemiological analysis comparing prevalence trends between MAPA with AECOSAN surveillance results that could lead to a change in the surveillance strategy to detect and correct any potential problem. In other countries, such as in Denmark, the joint analysis of the results from carcass swabs, serological surveillance of meat juices, ceacal samples and antibiotic treatment are performed by the Agriculture & Food Council to investigate any alerts deriving from any of the surveillance systems mentioned [21]. Even if the time of infection might cause mismatches among the results of these surveillance systems, since some detect antibodies and others are directed at isolating the bacteria, meaningful interpretations can be obtained over time. 

Early detection of human cases before they become an epidemic is crucial and the MSSSI, to combat *Salmonella* to decrease underreporting, has declared human salmonellosis notifiable since 2015. This means that now each human case that is detected (clinical signs + microbiological confirmation) must be notified to ISCIII, whether it leads to an outbreak. This way, there is now a complete database with information about occurrence of cases including location, symptomatology, and serovar involved. Prior to 2015, the only source of clinical information of the disease was the National Registry for Hospitalizations and data sources for microbiological information were scattered among different databases (regional surveillance, laboratory, and outbreaks databases). Surveillance systems implementation, however, requires a start-up period and a continuous evaluation of the quality of the system. Hospitalization registries have been used as a tool not only to evaluate the epidemiology of the most severe cases of *Salmonella* infections, but to support the evaluation of the quality and performance of the surveillance systems [22]. Regions now have the appropriate mechanisms to compile the cases’ microbiological and clinical information sent to the Notifiable Diseases system. Consequently, this will increase the quality of the information and simplify the information in a unique dataset covering all aspects of the disease. Also, the inclusion in the ND dataset of primary care cases’ clinical and demographic information will allow estimating a more realistic impact of the disease. This means that information on *Salmonella* human cases at national level should improve notably in the next years. Still, there is not an active surveillance system for human salmonellosis implemented in Spain as there is in other countries such as the United States (FoodNet), where laboratory, hospital and population surveys are conducted to estimate the burden and the attribution of foodborne illness [23].

Of the range of methods described in the literature to attribute the source to human salmonellosis [24], only outbreak investigation is carried out in Spain. This information is published in aggregated form. To be able to identify relevant risk factors for human infections, a closer collaboration between animal and human health stakeholders is required to access and analyze these data. Similarly, the information on pig status that is publicly available allows a basic analysis to detect temporal variations in the serovars present, but there is no further investigation to identify also a spatial variation, potential exposure risks for humans or quantitative microbial risk assessments to investigate the impact of different control scenarios and help decide the best strategy (e.g., reducing carcass load versus farm prevalence).

It is very likely that both the human and the animal monitoring of *Salmonella* would benefit from a better exchange of information and collaboration. For MSSSI, such exchanges could help to map the areas at higher risk of exposure, being able to alert the population to take measures to reduce that risk (e.g., enhance hygiene, cook thoroughly, etc.). For MAPA, it could improve the overall prevalence picture and help economizing routine surveillance efforts and expenses by identifying and targeting those areas from which more cases associated with pork meat arise.

## 5. Conclusions

There is ample room to improve the degree of collaboration between animal and human health surveillance on swine *Salmonella* in Spain and make the “One health” approach a reality, despite the huge amount of periodic detailed data collected separately in the sectors implied. A better collaboration among sectors along the different steps of the surveillance system would allow to estimate the impact of the infection more accurately, identify risk factors or detect spatio-temporal variations in the serovars present, to reach the overall objective of helping to reduce the human risk of exposure.

## Figures and Tables

**Figure 1 vetsci-06-00020-f001:**
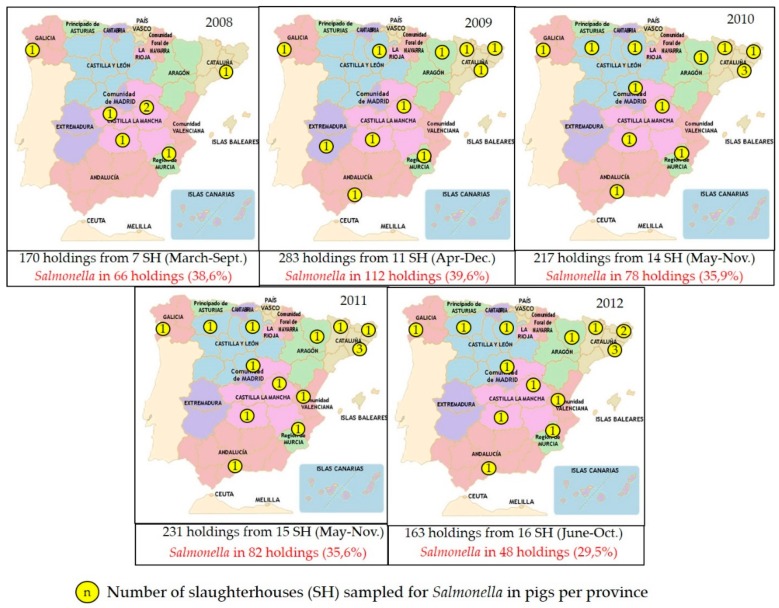
Number and distribution of slaughterhouses sampled and results of the *Salmonella* annual monitoring program in slaughter pigs in Spain. In black, number of total holdings and slaughterhouses (SH) sampled (time period of sampling). In red, positive (+) results to *Salmonella*. (Self-creation from the information retrieved from the reports in MAPA website).

**Figure 2 vetsci-06-00020-f002:**
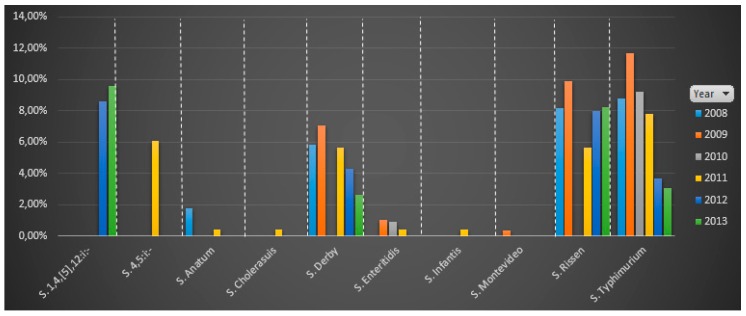
Distribution by serotype of the proportion of positive holdings to *Salmonella* (of total holdings tested per year) in slaughter pigs per year. This information is based on the annual Spain country reports (2008–2012).

**Figure 3 vetsci-06-00020-f003:**
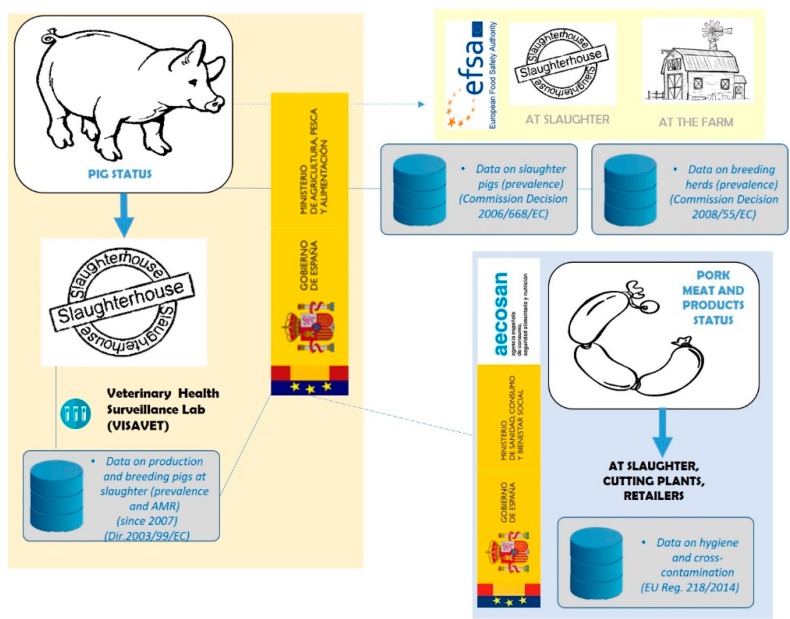
*Salmonella* surveillance “from farm to fork” (from pig source). AECOSAN notifies MAPA of the detection of meat contaminated with *Salmonella*, so that corrective action can be taken (indicated with a dotted line in Figure 1). In addition, the two surveys carried out by EFSA request are indicated, the results of which are kept by MAPA. AMR = antimicrobial resistance.

**Figure 4 vetsci-06-00020-f004:**
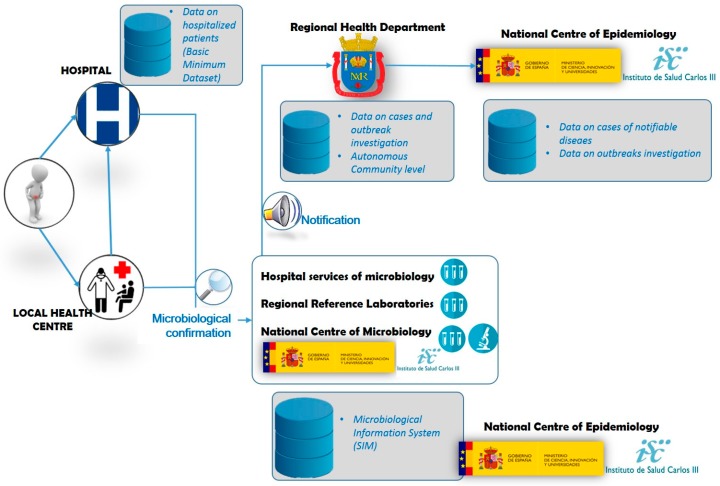
*Salmonella* surveillance in suspect patients in Spain (from ingestion of contaminated product or other exposure route). It summarizes the surveillance of *Salmonella* in humans, which starts with the onset of clinical signs in a suspect patient.

**Table 1 vetsci-06-00020-t001:** List of acronyms of institutions and systems used in this paper.

Acronym	Full Name
AECOSAN	Spanish Agency of Consumption, Food Security, and Nutrition [Agencia Española de Consumo, Seguridad Alimentaria y Nutrición]
ISCIII	Carlos III Institute of Health, which hosts the National Center of Epidemiology and the National Center of Microbiology. In this paper, ISCIII refers to the former, unless stated otherwise [Instituto de Salud Carlos III]
MAPA	Ministry of Agriculture [Mininsterio de Agricultura, Pesca y Alimentación]
MSSSI	Ministry of Health [Ministerio de Sanidad, Consumo y Bienestar Social]
RASFF	Rapid Alert System for Food and Feed (EU)
RENAVE	National Network of Epidemiological Surveillance [Red Nacional de Vigilancia Epidemiológica]
SCIRI	Coordinated System for the Exchange of Information (National) [Sistema Coordinado de Intercambio Rápido de Información]
SIM	Microbiological Information System [Sistema de Información Microbiológica]
VISAVET	VISAVET Health Surveillance Center [Centro de Vigilancia Sanitaria Veterinaria]

**Table 2 vetsci-06-00020-t002:** Institutional websites with information on swine-related *Salmonella* relevant for Spain.

Institutions	Websites URL (Last Visited: 22.11.2018)	Information Retrieved
EFSA	https://efsa.onlinelibrary.wiley.com https://www.efsa.europa.eu/en/topics/topic/salmonella	EU Summary reports on trends and sources of zoonoses, zoonotic agents and foodborne outbreaks (EFSA and ECDC) (2007–2012)
EFSA	https://www.efsa.europa.eu/en/efsajournal/pub/1377 https://www.efsa.europa.eu/en/efsajournal/pub/2329	Analysis of the baseline survey on the prevalence of *Salmonella* holdings with breeding pigs in the EU, 2008, Part A: prevalence estimates, Part B: factors associated with pen positivity
EFSA	https://www.efsa.europa.eu/en/efsajournal/pub/rn-135 https://www.efsa.europa.eu/en/efsajournal/pub/rn-206	Analysis of the baseline survey on the prevalence of *Salmonella* in slaughter pigs in the EU, 2006–2007, Part A: prevalence estimates, Part B: factors associated with lymph node positivity, surface contamination of carcasses, and serovar distribution
MAPA	https://www.mapa.gob.es/es/ganaderia/temas/sanidad-animal-higiene-ganadera/sanidad-animal/zoonosis-resistencias-antimicrobianas/zoonosis.aspx	Spain Reports by EFSA on zoonoses (2007–2012)
MSSSI	http://pestadistico.inteligenciadegestion.msssi.es/publicoSNS/comun/DefaultPublico.aspx	Hospital and Primary Health Centers Databases
ISCIII	http://www.isciii.es/isciii/es/contenidos/fd-servicios-cientifico-tecnicos/fd-vigilancias-alertas/fd-sistema-informacion-microbiologica/informes-generales.shtml	Annual SIM Reports (2014–2015)
ISCIII	https://publicaciones.isciii.es/unit.jsp?unitId=cne	Annual RENAVE reports (2014–2015)
AECOSAN	http://www.aecosan.msssi.gob.es/AECOSAN/web/seguridad_alimentaria/subseccion/SCIRI.htm	Annual SCIRI reports (2014–2015)

**Table 3 vetsci-06-00020-t003:** Relevant legislation for swine-related *Salmonella* surveillance.

Legislation	Subject	Reference
EU Legislation		
Commission Directive 2003/99/EC	Surveillance of *Salmonella* and other zoonotic agents	OJ, L325/31, 12.12.2003
Regulation 2160/2003	Control of *Salmonella* and other foodborne diseases	OJ, L325/1, 12.12.2003
Regulation 2073/2005	Microbiological criteria of food products	OJ, L338/1, 22.12.2005
Regulation 1441/2007	Microbiological criteria of food products (modification)	OJ, L322/12, 07.12.2007
Commission Decision 2006/668/EC	Design of *Salmonella* baseline survey in slaughter pigs	OJ, L275/51, 06.10.2006
Commission Decision 2008/55/EC	Design of *Salmonella* prevalence survey in breeding pigs	OJ, L14/10, 17.01.2008
Regulation 16/2011	Rapid alert system for food and feed (RASFF)	OJ, L6/7, 11.01.2011
Regulation 217/2014	Analysis of microbiological samples in carcasses	OJ, L69/93, 08.03.2014
Regulation 218/2014	Checks on the operator’s microbiological tests to control *Salmonella* through the food chain	OJ, L69/95, 08.03.2014
Spanish Legislation		
Royal Decree 2210/1995	National network of epidemiological surveillance	BOE 21, 2153-58, 24.01.1996
Royal Decree 1943/2004	Transposition of Directive 2003/99/EC	BOE 237, 32772-77, 01.10.2004
Order of the Ministry of Health [Sanidad y Consumo] SCO/3270/2006	National network of epidemiological surveillance for foodborne salmonellosis	BOE 255, 37238-9, 25.10.2006
Law 17/2011	Food security and national coordinated system for the rapid exchange of information (SCIRI)	BOE 160, 71283-319, 06.07.2011
Royal Decree 69/2015	Registry of the activity of primary health centers	BOE 35, 10789-809, 10.02.2015
Order of the Ministry of Health [Sanidad, Servicios Sociales e Igualdad] SSI/445/2015	Salmonellosis as a human notifiable disease	BOE 65, 24012-15, 17.03.2015

**Table 4 vetsci-06-00020-t004:** Overall biological risks and *Salmonella* spp. related notifications reported by SCIRI.

Notification Type	Notification Definition				Year			
					2014	2015	2016	Total
**Alert**	Overall food and drinks alerts				194	184	203	581
		Biological risk alerts related to animal products			44	42	53	139
			Biological risk alerts related to animal products with *Salmonella* spp. isolation		12	17	16	45
**Information**	Overall information notifications				1321	1333	1478	4132
		Biological risk information notifications			430	451	504	1385
			Biological risk information notifications related to animal Spanish products (Spanish involvement)		38	19	53	110
				Biological risk information notifications related to *Salmonella* spp. biological risk in animal Spanish products (Spanish involvement)	8	0	8	16
**Border products rejection**	Overall frontier products rejection				1296	1310	1078	3684
		Biological risk frontier products rejection related to animal products			242	242	154	638
			Frontier products rejection related to *Salmonella* spp. detection in animal products with *Salmonella* spp. isolation		6	1	1	8

**Table 5 vetsci-06-00020-t005:** Summary of results from the different sources, years 2014 and 2015.

Source	Institution	Cases Definition	Launch Date of the Dataset	Coverage	Year	
					2014	2015
**Notifiable Diseases**	ISCIII	Non-typhoid, non-paratyphoid salmonellosis	2015 (2014 volunteer notification)	National	7295	9069
**Outbreaks Database**	ISCIII	*Salmonella* confirmed outbreaks	2006	National	241 outbreaks (11.7% related to meat and derivates ^1^)	281 outbreaks (10.2% related to meat and derivates ^1^)
**SIM**	ISCIII	Any non-typhoid, non-paratyphoid *Salmonella* isolates	1995	72 labs from 11 regions	5001	5215
**Hospital registry** **(Minimum Basic Data Set)**	MSSSI	Hospitalizations in patients with gastroenteritis caused by *Salmonella* spp. (non-typhoid, non-paratyphoid)	1987	National	3526	3776

^1^ Percentage refers to those cases with information of the source of infection.

**Table 6 vetsci-06-00020-t006:** *Salmonella* spp. related notifications reported by RENAVE.

Year	Number of Cases with Available Serovar Information (% from Total Cases Reported)	*S. typhimurium*	*S. enteritidis*	*S. typhimurium* Monophasic	*S.* Newport	*Salmonella*, other Serotypes
**2014**	3877 (50.5%)	1640 (44.5%)	1220 (33.1%)	83 (2.3%)	17 (0.46%)	69 (1.87%)
**2015**	5299 (58.4%)	2152 (61.1%)	1066 (30.3%)	153 (4.3%)	28 (0.79%)	122 (3.5%)

**Table 7 vetsci-06-00020-t007:** Summary of results from the different sources by region. SIM = Microbiology Information System; MBDS = Minimum Basic Data Set.

Region	2014			2015		
	Cases (Notifiable Diseases)	Isolates (SIM)	Hospitalizations (MBDS)	Cases (Notifiable Diseases)	Isolates (SIM)	Hospitalizations (MBDS)
Andalusia			523			503
Aragon	442	461	194	555	557	190
Asturias	437	442	132	376	380	113
Balearic Islands			69			89
Canary Islands	568	575	130	370	378	83
Cantabria			44	83		38
Castile-La Mancha	149	150	178	177	177	210
Castile and Leon	693	457	339	811	436	367
Catalonia	1778	1806	489	1968	2090	522
Valencian Community	2217		385	2539		422
Extremadura	79	81	81	287	66	97
Galicia			151			197
Madrid			431	778		547
Murcia			143			153
Navarra	280	282	52	337	327	49
Basque Country	432	533	147	625	616	135
La Rioja	185	189	24	148	149	33
Ceuta	24	25	9	3	39	17
Melilla	11		5	12		11
**Total**	**7295**	**5001**	**3526**	**9069**	**5215**	**3776**

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
