# Peer review of "Salmonella Surveillance Systems in Swine and Humans in Spain: A Review"

_vetsci, 2019, doi:10.3390/vetsci6010020_

Round 1

Reviewer 1 Report

The manuscript is significantly improved over the first version, and includes the alterations advised by myself where applicable.

There remain a few minor changes needed or suggestions, viz:

Advise Redraft Table 3 so it is not two separate sub-tables. Suggest indented legislation headings as sub-headings of ‘EU Legislation’ and ‘Spanish legislation’ in 1st column.

Line 238: 'deduce' not 'deduct'

Line 262: 'vomiting' not 'vomits'

Lines 280 + 282: – Actually Table 5, not Table 1. Amend other cross-references in the text too

Lines 285, 287: Actually Table 6 not Table 2. Amend other cross-references in the text too

Line 287: “% from the overall cases with information” It is still unclear what this means.

Line 466: This is published in EFSA Supporting Publications – the reference formatting is incorrect

Author Response

The authors wish to thank Reviewer 1 for the comments received. The minor changes suggested have been taken into account and are now corrected according to Reviewer 1 comments (highlighted in yellow in the manuscript):

-          Table 3 has been redrafted as a single table with indented subheadings in the first column corresponding to EU Legislation and Spanish Legislation

-          Line 238: “deduct” has been replaced by “deduce”

-          Line 261: “vomits” has been replaced by “vomiting”

-          Cross-references in the text about Table 5 and 6. There are no such tables in this manuscript. Only Tables 1 to 4 exist. Cross-references have been checked in the text.

-          Line 291: “% from the overall cases with information”. This is only applicable to the first column that gathers the number of cases with available serovar information. The text in the table has been amended accordingly.

-          Line 466: the EFSA Supporting Publication reference has been corrected.

Reviewer 2 Report

Did you find any information from published articles as searched from the PubMed? If yes, I think details of the number of article included and details information in necessary to improve the manuscript.

Author Response

The authors have taken into account the suggestion made by Reviewer 2 and have included in line 163 a new paragraph with the findings from the literature search, together with an annex with the full references.